# Research Progress on the Development of Porcine Reproductive and Respiratory Syndrome Vaccines

**DOI:** 10.3390/vetsci10080491

**Published:** 2023-07-28

**Authors:** Hang Zhang, Qin Luo, Yingxin He, Yajie Zheng, Huiyang Sha, Gan Li, Weili Kong, Jiedan Liao, Mengmeng Zhao

**Affiliations:** 1School of Life Science and Engineering, Foshan University, Foshan 528000, China; 2112159017@stu.fosu.edu.cn (H.Z.); luoqin121104@163.com (Q.L.); hyxin@outlook.com (Y.H.); zhengyajie2022@163.com (Y.Z.); huiyangsha2022@163.com (H.S.); ligan1227@163.com (G.L.); 2Gladstone Institutes of Virology and Immunology, University of California, San Francisco, CA 94158, USA; weili.kong@gladstone.ucsf.edu

**Keywords:** porcine reproductive and respiratory syndrome, vaccine, advantages and disadvantages, applicability, research progress

## Abstract

**Simple Summary:**

Vaccine immunization measures are mainly adopted for porcine reproductive and respiratory syndrome (PRRS) prevention. We summarize the advantages, disadvantages, and applicability of currently developed PRRSV vaccines, including modified-live virus, inactivated, recombinant subunit, live vector, DNA, gene-deletion, synthetic peptide, and virus-like particle vaccines, as well as various other vaccines. These results provide a theoretical foundation for the development of new vaccines.

**Abstract:**

Porcine reproductive and respiratory syndrome (PRRS) is a highly contagious disease in the pig industry, but its pathogenesis is not yet fully understood. The disease is caused by the PRRS virus (PRRSV), which primarily infects porcine alveolar macrophages and disrupts the immune system. Unfortunately, there is no specific drug to cure PRRS, so vaccination is crucial for controlling the disease. There are various types of single and combined vaccines available, including live, inactivated, subunit, DNA, and vector vaccines. Among them, live vaccines provide better protection, but cross-protection is weak. Inactivated vaccines are safe but have poor immune efficacy. Subunit vaccines can be used in the third trimester of pregnancy, and DNA vaccines can enhance the protective effect of live vaccines. However, vector vaccines only confer partial protection and have not been widely used in practice. A PRRS vaccine that meets new-generation international standards is still needed. This manuscript provides a comprehensive review of the advantages, disadvantages, and applicability of live-attenuated, inactivated, subunit, live vector, DNA, gene-deletion, synthetic peptide, virus-like particle, and other types of vaccines for the prevention and control of PRRS. The aim is to provide a theoretical basis for vaccine research and development.

## 1. Introduction

Porcine reproductive and respiratory syndrome (PRRS) is a highly contagious and virulent infectious disease caused by the PRRS virus (PRRSV). It can cause reproductive disorders, such as abortion, premature birth, and mummified fetuses, as well as respiratory diseases in pigs of various ages [1]. PRRSV infection in pigs can lead to immunosuppression, impair immune function, and cause significant economic losses to the pig industry [2]. PRRS was first discovered in North Carolina in 1987 [3], and since then, various strains have been isolated worldwide. In 1991 and 1992, Lelystat virus and VR2332 strains were isolated from infected pigs in Europe and America [4,5]. PRRSV LV and VR2332 strains were genetically classified as PRRSV type 1 (PRRSV-1) and type 2 (PRRSV-2). In the recent taxonomic classification, PRRSV-1 and PRRSV-2 have been assigned to species Betaarterivirus suid 1 and Betaarterivirus suid 2, respectively. PRRSV-2 strains are mainly prevalent in China, where highly pathogenic PRRS (HP-PRRS) outbreaks have occurred with high fatality rates in piglets [6,7]. In 2012, a highly variable NADC30-like strain was identified in China [8], which can easily recombine with other strains. Additionally, the NADC34-like PRRSV is prevalent in many regions of China [9,10]. PRRSV continues to mutate and recombine, and current vaccines cannot provide adequate protection, making prevention and control more complex.

PRRSV is a positive-sense single-stranded RNA virus with an envelope and belongs to the genus Arterivirus. Its genome is approximately 15 kb in length, consisting of 11 open reading frames (ORFs), including ORF1a, ORF1b, ORF2a, ORF2b, ORF3-7, ORF5a, and ORF1aTF, which overlaps with the nonstructural protein (NSP) 2-encoding region of ORF1a [11]. Among them, ORF1a, ORF1aTF, and ORF1b account for 80% of the total genome. The translated polypeptides pp1a and pp1ab are cleaved into 15 NSPs, including NSP1α, NSP1β, NSP2-related proteins (NSP2N, NSP2TF, and NSP2), and NSP3–NSP12, which are involved in viral replication [12,13]. ORF2a, ORF2b, ORF3, ORF4, ORF5, and ORF5a encode six envelope-related proteins, namely, GP2a, GP2b, GP3, GP4, GP5, and GP5a, respectively. The nonglycosylated matrix (M) protein encoded by ORF6 and the nucleocapsid (N) protein encoded by ORF7 are dominant structural proteins with strong immunogenicity. The GP5 protein is a major structural protein with good immunogenicity, capable of inducing the production of neutralizing antibodies (NAs) by the body, and plays a crucial role in persistent viral infection [14]. Therefore, GP5 is still the preferred protein in PRRS vaccine research [15,16].

Antibody-dependent enhancement (ADE) is a significant challenge to PRRS vaccine development, where preexisting non-neutralizing or sub-neutralizing antibodies promote virus entry and replication. The first report of ADE with PRRSV infection dates back to 1993 [17]. A study found that virus replication was increased in fetuses inoculated with the virus plus antibodies compared to those inoculated with the virus only [18]. Yoon et al., later reported that pigs injected with a sub-neutral amount of PRRSV-specific IgG before virus infection had increased viremia, and PRRSV-specific IgG increased the production of escape mutants of the virus [19]. Fcγ receptors can participate in ADE caused by PRRSV [20,21]. However, some studies have found that sub-neutralized IgG does not cause ADE in vivo, and modified-live PRRSV vaccines can provide partial cross-protection against heterologous strains [22,23,24]. At present, in vivo experiments are insufficient to confirm whether ADE plays a crucial role in PRRSV pathogenesis. ADE upon infection with PRRSV may vary with different strains and conditions. In vitro studies have shown that anti-PRRSV serum has the most potent effect on infection at a 27-fold dilution. The sera from one pig with an S/P ratio of 1.2 and a serum NA titer of 1/5.9 were used in the ADE test [25]. After the passive transfer of PRRSV NAs, a higher concentration of the PRRSV NA titer (1:32) in serum can induce complete protection. However, this is only applicable to some young pigs, and the titer (1:8) cannot prevent PRRSV replication in the lungs or transmission to other peripheral lymphoid tissues [26]. The results of these studies are contradictory, and further research is needed to determine the potential mechanism of ADE during PRRSV infection. It is also necessary to determine whether the attenuated PRRS vaccine exacerbates the disease in pigs infected with a new virus strain.

Currently, vaccination is the primary measure for PRRS prevention. Domestic inactivated vaccines are derived from the CH-1a classical strain, while attenuated vaccines originate from different sources and are divided into classical strains (VR-2332, CH-1R, and R98) and highly pathogenic strains (JXA1, TJ, HuN4, and GD), which are widely used. The popular PRRSV in China can be divided into four lineages, namely, lineages 1, 3, 5, and 8 [27,28]. Since the emergence of PRRSV in China, lineage 8 has dominated, including the classic PRRSV strain (CH-1a-like) that circulated before 2006 and the HP-PRRSSV-like strain that circulated after 2006 [29]. Lineage 1, also known as the NADC30-like strain, has been rapidly spreading nationwide since 2013, and the current clinical detection rate is comparable to that of lineage 8 (HP-PRRSV-like) [30]. Although lineage 5 (BJ-4-like/VR2332-like) appeared as early as 1996, it has been in a non-pandemic state in China with a low clinical detection rate.

Compared with lineage 8 (CH-1a-like) and lineage 5, HP-PRRSV-like and NADC30-like strains showed higher genetic variation and recombination rates. These characteristics may render current vaccines ineffective, making it easier for them to evade immune surveillance [31]. Additionally, the limitations of current PRRS vaccines extend beyond their inability to protect against heterologous strains. They also fail to meet the swine industry’s need for comprehensive and universal protection and carry the risk of reversion to virulence. Although commercial vaccines provide reasonable protection against homologous strains in virology and clinical settings, they are not foolproof. Continuing research and development of genetically engineered vaccines and marker vaccines can address the limitations of current PRRS vaccines. This manuscript reviews the advantages, disadvantages, and applicability of various PRRS vaccines, including modified-live virus (MLV), inactivated, subunit, live vector, DNA, gene-deletion, synthetic peptide, and virus-like particle (VLP) vaccines, as well as various other vaccines. The goal is to provide a theoretical foundation for the development of new vaccines that can offer enhanced protection against PRRSV in regions affected by this disease. In the context of next-generation vaccine development, further evaluation of the screening method of the proteome full-domain peptide library and the identified antigens will play an important role in developing new vaccines that can better protect against PRRSV in China, particularly against heterologous strains and strains with higher genetic variation and recombination rates.

## 2. MLV Vaccines

Since the discovery of PRRSV, several MLV vaccines have been developed to combat PRRSV-1 and PRRSV-2. Porcilis PRRS (Merck, Madison, NJ, USA), UNISTRAIN PRRS (Hipra, S.A., Amer, Spain), Ingelvac PRRSFLEX EU (Boehringer Ingelheim, Ingelheim am Rhein, Germany), ReproCyc PRRS EU (Boehringer Ingelheim, Ingelheim am Rhein, Germany), and Suvaxyn PRRS MLV (Zoetis, Belgium, WI, USA), developed for PRRSV-1, were approved for marketing in 2000, 2013, 2015, 2015, and 2018, respectively, and they are widely used in Western European countries. The Ingelvac PRRS MLV (Boehringer Ingelheim, Ingelheim am Rhein, Germany), Fostera PRRS (Zoetis, Belgium, WI, USA), PrimePac PRRS (Merck, Madison, NJ, USA), and Prevacent PRRS (Eli Lilly, Ind, USA) vaccines developed for PRRSV-2 were approved for marketing in 1994, 2012, 2014, and 2018, respectively, and are primarily used in the United States. Details of vaccine efficacies and immunity induced by the vaccines are described in Table 1.

In China, PRRS vaccines have mainly been developed for PRRSV-2, and there are seven MLV vaccines, including Ingelvac PRRS, CH-1R, R98, JXA1-R, TJM-F92, HuN4-F1, and GDr-180 (Table 2). However, no vaccine can claim to completely prevent the disease [32]. PRRSV, whether the wild type or attenuated, induces low levels of cellular immunity, and NAs are not produced until the later stages of infection. Studies have shown that after vaccination, pigs developed viremia but an undetectable level of NAs and reduced frequencies of virus-specific gamma-interferon-secreting cells (IFN-γ-SC). The levels of interleukin-10 (IL-10) produced by peripheral blood mononuclear cells (PBMCs) were inversely correlated with IFN-γ responses. These results indicate that IFN-γ and IL-10 play an important role in the development of immunity against PRRSV. The protective effect of MLV on pigs was not based on humoral factors, but on cell-mediated immunity [33]. Overall, available evidence suggests that all previously approved PRRS MLV vaccines cause weak humoral and cellular immune responses.

**Table 1 vetsci-10-00491-t001:** Foreign MLV vaccine statistics.

Type	Vaccine Strain Name	Approval Time	Source of Vaccine Strains	Company	Vaccine Efficacies	Immune Responses	References
PRRSV-1	Porcilis PRRS	2000	DV	Merck	Minor local reactions	Humoral and cellular immune responses	[34]
	UNISTRAIN PRRS	2013	VP- 046 BIS	Hipra	The mean duration of fever was shortened and the number of fever days was significantly lower	Humoral immune response	[35]
	Ingelvac PRRSFLEX EU	2015	94811	Boehringer Ingelheim	Increases in Average Daily Weight Gain; reduces the appearance of clinical signs	Humoral and cellular immune responses	[36]
	ReproCyc PRRS EU	2015	94881	Boehringer Ingelheim	Reduces viremia	Humoral immune response	[37]
	Suvaxyn PRRS MLV	2018	96V198	Zoetis	No local reactions	Humoral and cellular immune responses	[38]
PRRSV-2	Ingelvac PRRS MLV	1994	ML (VR2332)	Boehringer Ingelheim	Reduce viremia and lung lesions	Cellular immune responses	[39]
	Fostera PRRS	2012	/	Zoetis	Reduces viremia and lung lesions	Humoral and cellular immune responses	[40]
	PrimePac PRRS	2014	/	Merck	Reduces rectal temperatures, clinical respiratory scores, lung lesion scores, and levels of virus load in serum and lung tissue	Cellular immune responses	[41]
	Prevacent PRRS	2018	/	Eli Lilly	/	/	Not reported

Experiments evaluating vaccine efficacy have shown that PRRSV-MLVs provide adequate protection against genetically homologous wild-type PRRSV strains but provide only partial or no protection against heterologous strains [42]. In 2012, research indicated that vaccinated pigs had significantly lower mortality rates, fewer days of fever, a lower incidence of catarrhal bronchopneumonia, higher weight gain, and lower levels of viremia compared to pigs that were not vaccinated with MLV. The study found that immunization with a PRRSV vaccine with attenuated PRRSV-1 provides partial protection against attacks from highly virulent PRRSV-2 strains [43]. In another experiment testing the efficacy of the commercially modified PRRSV-1 subtype live vaccine UNISTRAIN, Bonckaert et al. [36] found that the average fever time of vaccinated pigs was shortened, and the number of fever days was significantly lower than that of control pigs. A few vaccinated animals experienced fever, respiratory disorders, and conjunctivitis. The average viral titer in nasal secretions after stimulation in the vaccination group was significantly lower than that in the control group. Compared to the control group, the duration of viremia in the vaccine group showed a slight decrease, although the difference was not statistically significant. Recent studies have shown that MLV vaccines against PRRSV-1 protect 1-day-old piglets from heterologous PRRSV-1 infection at weaning [44]. Further, MLV vaccines against PRRSV-2 have been shown to offer cross-protection against heterologous PRRSV-1 in pregnant gilts throughout their trimester [45,46,47]. They also play a protective role against HP-PRRSV infections [41]. However, MLV vaccines against PRRSV-1 can only protect gilts from heterologous PRRSV-1, and not PRRSV-2, in the third trimester [48]. Maragkakis et al. [49] conducted a study wherein they observed that the intradermal (ID) injection of the PRRSV-1 MLV vaccine in piglets had a beneficial effect on their health and production performance, as opposed to intramuscular (IM) inoculation. This positive impact was evidenced by an augmentation in body weight at slaughter and a reduction in lung and pleurisy lesion scores. Therefore, all MLVs currently available do not provide comprehensive protection and cannot fully meet the needs of the swine industry.

The challenge of safety arises for PRRSV-MLVs due to vertical transmission, a decrease in the number of piglets born alive, and the potential to cause persistent infections in vaccinated hosts. Charerntantanakul and Wang et al., demonstrated that pigs vaccinated with MLV could develop viremia for up to four weeks after immunization, leading to the transmission of the vaccine virus to unvaccinated pigs [42,50]. Therefore, the safety of PRRSV-MLVs should be considered. Since the approval of the Ingelvac PRRS MLV vaccine for marketing in 1994, it has been widely used in China and the United States for the vaccination of pig farms, and viruses causing subsequent outbreaks of PRRS on these farms were found to have nearly identical nucleotide sequences to those of vaccine strains reported in both countries [51]. In addition, recombination between MLVs and wild-type strains has exacerbated PRRS outbreaks [52,53]. Recent research has revealed that a new strain in Denmark resulted from a recombination event involving the Amervac strain (Unistrain PRRS vaccine; Hipra) and the 96V198 strain (Suvaxyn PRRS; Zoetis AH). The 96V198 strain primarily contributed to the genetic makeup of the strain, encompassing ORFs 1-2 and a portion of ORF 3. On the other hand, the Amervac strain acted as the minor parent, contributing to the remaining portion of the genome [54]. Therefore, research on new PRRSV-MLVs should lead to breakthroughs in vaccine efficacy and safety.

Clinically, MLV vaccines are usually used on farms that have already had PRRS outbreaks. For farms where PRRS has not occurred, MLV vaccines are usually used for the prevention of disease, but not for the prevention of infection. After vaccination, the attenuated strain in the vaccine will multiply in the pig, stimulating the body to produce antibodies to fight the disease [55]. MLVs offer suitable immune prevention in pigs aged 3–18 weeks. Further, they can exert protective effects within 7 days after vaccination, and the protection period is longer, reaching more than 16 weeks. The immunization of sows or replacement gilts can also enhance their reproductive performance. The vaccine strain determines the effectiveness of the attenuated vaccine, and PRRSV genetic diversity influences the effectiveness of the attenuated vaccine [56,57]. Moreover, the vaccine has a good preventive effect on homologous strains. MLVs have a specific effect on preventing PRRS; however, vaccine strains should be carefully selected before use because of potential safety concerns.

## 3. Inactivated Vaccine

Compared to MLV vaccines, inactivated PRRSV vaccines have been licensed worldwide owing to their improved safety. There are many commercially available inactivated vaccines targeting PRRSV-2 strains (CH-1a, ISU-P, JXA1, VD-A1, and SD-1) and PRRSV-1 strains (LV, VD-E1, VD-E2, and PRRS-CY-218-JPD-P5-6-91 or Olot) in China and other countries. However, since 2005, the United States has no longer provided these vaccines because of their observed poor efficacy [42]. Opriessnig et al., confirmed that an inactivated PRRSV vaccine could shorten the duration of virus persistence in boar semen after infection [56]. However, Nilubol immunized boars with an inactivated vaccine from the same source (PRRomisee, Intervir, VR-2402 vaccine strain) 2 years later but failed to show a reduction in the virus content in semen [57]. The difference between the two tests is that the infection strain used in the former study was the same as the vaccine strain (VR-2402), and the infection strain used in the latter study was VR-2385, which indicates that cross-protection induced by the inactivated vaccine is weak. At the same time, some reports have explained the adverse effects of an inactivated PRRSV vaccine on wild virus infection, which is manifested by the lack of detectable PRRSV-specific antibody production [58] and the lack of a cell-mediated immune (CMI) response [59,60]. In the PRRSV-specific immune response (KV/ADJ, Progressis, Merial Labs, PRRSV-1 and PRRomisee, Intervet, PRRSV-2) induced by two genotypes of an inactivated PRRSV vaccine, the titer of viral NAs is generally lower than eight, which cannot effectively clear PRRSV [33]. Therefore, in in vivo experiments, the inactivated PRRSV vaccine failed to result in a statistically significant difference among pigs infected with wild-type PRRSV [61].

Inactivated PRRSV vaccines have the advantages of safety, convenient storage, and ability to be transported, and in general, maternal antibody interference in an inactivated vaccine is relatively weak [55]. In addition, the number of stillbirths and the abortion rate of sows vaccinated with inactivated vaccines are decreased significantly, and the fertility of sows and survival rate of piglets are increased [58]. However, inactivated vaccines also have certain disadvantages, such as poor immune effects on heterologous strains, large vaccination doses, multiple immunization times, high costs, and a specific time frame required to produce immunity. The inactivated oil-adjuvant vaccine made from the isolated virus strain (CH-1a strain) in China has a good immune-modulating effect and is maintained for a long time [62]. Only 20 days after the second immunization can pigs obtain immunity, which is vital. It also results in good protection for more than 80% of vaccinated pigs and can impart a period of immunity of up to 6 months. The inactivated oil-adjuvant vaccine of the wild-type virus strain (SI strain) is suitable for piglet immunization [63]. Once the animal is immunized, it can confer more than 6 months of protection. At the same time, the vaccine can also be used for the vaccination of replacement gilts and boars. The successful implementation of Hungary’s PRRS Eradication Programme has been substantiated by recent research. This achievement was accomplished by administering an inactivated vaccine to sows and adopting segregated rearing for their offspring. Notably, the production remained nearly uninterrupted throughout the entirety of the population replacement process [64]. Several studies have indicated that the utilization of a combination of the MLV vaccine and inactivated vaccine provides enhanced protection against PRRSV infection in pigs. The administration of the MLV vaccine during the eighth week of gestation in sows, followed by the re-inoculation of the commercial inactivated vaccine three weeks prior to delivery, effectively diminishes the occurrence rate of PRRSV. Furthermore, it has been noted that weaned piglets exhibit a greater proportion of PRRSV-seropositive individuals [65]. Furthermore, the combination of the MLV vaccine and inactivated vaccine can provide partial protection to piglets, reducing pathological lung loss, improving weight gain, and decreasing viremia. This effect may be attributed to the production of NAs in sows induced by the inactivated vaccine [66]. In conclusion, these data indicate that inactivated PRRSV vaccines might have a potential therapeutic role in treating PRRS, rather than preventing disease. Hence, before developing new formulations containing adjuvants, further exploration of the therapeutic benefits of inactivated vaccines may be warranted.

## 4. Subunit Vaccine

The subunit vaccine, also known as a biosynthetic subunit vaccine, is made by inserting specific-antigen-encoding genes into viruses or cells that are prone to proliferation to express effective specific antigens and purifying them [67]. Although they are safe and stable, their immunogenicity is usually low. Immunogenic PRRSV proteins exist in both structural and nonstructural forms. The virus has six membrane proteins, labeled major (GP5 and M) and minor (GP2a, E, GP3, and GP4) envelope proteins [68]. GP5 is the most abundant glycoprotein and the main inducer of NAs, but it varies greatly among PRRSV strains [16,69]. The M protein is a highly conserved structural protein, and it is nonglycosylated and helps in virus assembly and budding. GP5 and M exist in a “heterodimeric complex” linked by covalent disulfide bonds in cells infected with the virus. The N protein is the most abundant in virus-infected cells [70], accounting for about 40% of the virions. The minor envelope proteins associate through non-covalent interactions to form a “heterotetrameric complex” in infected cells [71]. Apart from GP5, minor proteins also possess B-cell epitopes that are virus-neutralizing. The potential T-cell epitopes in the PRRSV exist in proteins GP3, GP4, GP5, M, N, NSP2, NSP5, NSP9, and NSP10. In order to induce effective anti-PRRSV immunity, it is crucial to expose the immunogenic viral proteins in the form of “heterodimeric and heterotetrameric complexes” for the efficient induction of specific and protective B- and T-cell responses in innovative vaccine formulations. Shortly after the discovery of PRRSV, the PRRSV structural protein, expressed by a baculovirus, was tested as a potential subunit vaccine [72]. In the earliest study, Plana et al., carried out an animal immunity test by constructing PRRSV ORF2–ORF7 expression vectors. The results showed that GP3 and GP5 could provide specific protective effects, which suggested that GP3 and GP5 subunit vaccines have good immunogenicity and that these could be used as candidate genes for developing recombinant subunit vaccines [73]. Since then, a transgenic plant oral subunit vaccine against PRRSV has also been tested [74]. However, the experimental plant-based vaccine has the same defect as the baculovirus-based subunit vaccine; specifically, its efficacy in pigs is limited [75]. Song et al. [76] amplified a truncated GP5 gene missing its signal peptide and transmembrane sequence via overlapping PCR and inserted it into the prokaryotic expression vector pET-32a or pGEX-6p-1, and His or GST tags were added to induce expression and facilitate purification, and animal experiments were performed. The results showed that flagellin from Salmonella typhimurium was an effective adjuvant, as it increased the induction of anti-GP5 antibodies and induced humoral and cellular immune responses. However, this study performed an animal immunization experiment in a mouse model, and the efficacy of the vaccine in pigs was not fully understood. Hu et al. [77] genetically engineered maize calli to produce the PRRSV M protein and induced serum and intestinal mucosal antigen-specific antibodies in mice via the oral administration of transgenic plant tissue. The results showed that transgenic maize plants are an effective way to produce subunit vaccines to generate systemic and mucosal immune responses to PRRSV. An et al. [78] successfully expressed an antigen protein derived from PRRSV in Arabidopsis plants, realizing the successful production of a recombinant protein vaccine. Peng et al. [79] compared the immunopotentiation of four natural adjuvants to improve the efficacy of the PRRS subunit vaccine and found that the astragalus/bacillus adjuvant resulted in the most effective immunopotentiation of the PRRSV GP5 subunit. Peng et al. [80] also studied the immune-enhancing effect of Taishan Pinus massoniana pollen polysaccharide (TPPPS) and Freund’s adjuvant on the PRRSV subunit vaccine. Based on the results, it was expected that a moderate dose of TPPPS as a GP5 adjuvant could become a candidate PRRSV subunit vaccine.

Compared to traditional vaccines, subunit vaccines do not contain nucleic acid substances; therefore, they are safer. The immunization of animals does not result in persistent infection, and the immune response can be distinguished from that to wild virus infection. Some studies have shown that vaccination in late-term pregnancy with a PRRSV subunit vaccine is efficacious against reproductive failure caused by heterogeneous PRRSV-1 and PRRSV-2 infection [81]. This is conducive to the prevention and control of various diseases. However, its production cost is high, and its immunogenicity might be better than that of attenuated or inactivated vaccines; therefore, its development time is relatively long. In the long run, the PRRSV subunit vaccine has broad application prospects for the prevention and control of PRRS. However, the duration of immunity is short, and it must be used in combination with an adjuvant. Adjuvants can promote the response of T cells or B cells by enhancing the activity of macrophages participating in the immune response to antigens. It has been found that a variety of cytokines can enhance the specific immune response to the vaccine [82]; therefore, developing adjuvants with beneficial effects will help to enhance the specific immune response to PRRSV subunit vaccines.

## 5. Live Vector Vaccines

A live vector vaccine is prepared by transforming a recombinant PRRSV antigen gene into a live vector capable of expressing exogenous viral proteins using genetic engineering technology. The vaccine can elicit specific immune responses after the direct immunization of animals [83]. Live PRRSV vector vaccines mainly use poxvirus, adenovirus, herpes virus, and porcine pseudorabies virus (PRV) as vectors to express the central PRRSV immunogenic genes [84]. Alonso et al. [85] successfully constructed a recombinant transmissible gastroenteritis virus expressing the GP5 protein and detected high levels of a GP5-specific antibody. Fang et al. [86] constructed a recombinant PRV vector expressing PRRSV GP5 and M proteins. The results showed that the recombinant virus vector vaccine could induce the production of NAs and stimulated lymphocyte proliferation. Tian et al. [87], using PRRSV (DS722 strain) as a live viral vector, constructed a multicomponent vaccine virus, DS722-SIV-PCV2, expressing the protective antigens of swine influenza virus (SIV) and porcine circovirus type 2 (PCV-2). The test results showed that DS722-SIV-PCV2 has potential as a candidate trivalent vaccine and also revealed its possibility for use against PRRSV as a potential live virus vaccine vector. Recent studies have shown that rPRRSV-E2, a candidate vaccine for a recombinant PRRSV vector expressing the classical swine fever virus (CSFV) E2 protein, is necessary for the effective prevention and control of HP-PRRSV and CSFV [88,89]. In recent years, live viral vectors have shown great potential in PRRSV vaccine research, but their production costs are high, their efficiency is low, and the carrier virus may also trigger an immune response, thereby reducing the effectiveness of the vaccine. Although researchers still do not understand the natural occurrence of the virus, there are doubts about whether recombination will occur between a live virus vector vaccine strain and wild or attenuated virus vaccine strains. Nevertheless, we believe that with the development of biotechnology, live viral vector vaccines have the potential to become the focus of the PRRSV vaccine field.

## 6. DNA Vaccines

DNA vaccines, also known as nucleic acids or gene vaccines, are based on the direct introduction of eukaryotic expression vectors, producing specific antigens of pathogenic microorganisms, into the body of an animal to stimulate the production of a typical immune response [90]. Kwang et al. [91] constructed four recombinant DNA vaccines, PRRSV ORF4–ORF7, and animal immunity tests were performed. PRRSV-specific antibodies were detected in 71% of immunized animals through ELISA, virus neutralization, and Western blotting analysis. In addition, cellular immune responses were detected in 86% of immunized pigs through IFN-γ and/or proliferation assays. The results confirmed that the antigen-neutralizing epitopes of PRRSV were mainly concentrated in the GP4 and GP5 proteins and showed that the DNA vaccine could induce humoral and cellular immunity. Siriseewan et al., found that pre-immunization with a DNA vaccine encoding a truncated PRRSV N protein, administered 2 weeks before MLV immunization, could improve PRRSV-specific immunity, as observed by increased NA titers and PRRSV-specific IFN-γ production and reduced IL-10 and PRRSV-specific Treg production [92]. A DNA vaccine constructed with the PRRSV GP5-Mosaic sequence was also complexed with cationic liposomes and intradermally and intramuscularly injected into experimental pigs. The results showed that in pigs inoculated with the GP5-Mosaic vaccine, IFN-γ mRNA expression in peripheral blood mononuclear cells was significantly higher than that in the control group [93]. A “suicide” DNA vaccine is a new type of DNA vaccine developed based on a virus replicon. Sun et al. [94] constructed the “suicide” DNA vaccine pSFV1CS-E2 with the PRRSV GP5 and CSFV E2 genes. The results showed that the “suicide” DNA vaccine pSFV1CS-E2, co-expressing GP5 and E2, could induce specific humoral and cell-mediated immune responses to PRRSV and CSFV. Jiang et al. [95] reported the typical immune response associated with PSFV-ORF5m/ORF6 by constructing a GP5m and PRRSV M protein-co-expressing “suicide” DNA vaccine, namely, PSFV-ORF5m/ORF6. This experiment also provided new ideas and ways to develop a PRRSV DNA vaccine. Compared with traditional vaccines, DNA vaccines are safer and have a simpler preparation process; the production cycle is short, but DNA-based vaccines have the same drawbacks as other methods, such as low immunogenicity and insufficient resolution of the heterogeneity of PRRSV. “Suicide” DNA vaccines represent a significant breakthrough in the development of DNA vaccines due to their safety, efficiency, and ease of preparation. The low expression of proteins can be attributed to the inadequate uptake of DNA vaccine plasmids by antigen-presenting cells. Researchers have explored alternative routes and methods for delivering DNA vaccines to enhance their uptake [96]. Subcutaneous injection needles have been studied as a replacement for the usual subcutaneous and intramuscular injection pathways. Other methods, such as transdermal, mucosal, and microneedle applications, as well as jet injection and electroporation, have also been tested to improve the immunogenicity of DNA vaccines [96].

## 7. Gene-Deletion Vaccines

Gene-deletion vaccines are constructed by excising the virulence-related genes of virulent strains using genetic engineering technology [97]. Leng et al. [98] obtained a PRRSV gene-deletion vaccine strain, TJM-F92, which could be inoculated during any growth phase of pigs, with an 80% protection rate. Xu et al. [72,99] constructed an infectious cDNA clone by deleting 75 nucleotides in the NSP2 region of the attenuated vaccine HuN4-F112 strain. An immune-dominant B-cell epitope gene fragment encoding Newcastle disease virus nucleoprotein (NP) was inserted into the deletion site. Piglets were immunized with recombinant PRRSV, followed by a challenge, and the vaccinated piglets produced specific antibodies to both NP and PRRSV, but no antibodies against NSP2. No immunized piglets in the experimental group showed clinical signs after the challenge, whereas all in the control group died 10 days after the challenge. The results showed that recombinant PRRSV rHuN4-F112-Δ508-532 could be used as a potential marker vaccine for PRRS. Hu et al. [100] constructed a live Japanese encephalitis virus (JEV) vaccine, SA14-14-2, with a CMV promoter and inserted PRRSV GP5/M into the deletion site of the replicon, based on JEV DNA, to develop the Balb/c chimeric replicon vaccine candidates pJEV-REP-G-2A-M-IRES and pJEV-REP-G-2A-M for animal immunoassays. ELISA data analysis showed that GP5/M replicons induce better immune responses; therefore, GP5/M JEV DNA-based replicons could be further developed into a new and safe PRRSV vaccine candidate. Zhou et al. [101] sequenced 61 PRRSV NSP2 hypervariable regions (NSP2HVs) and ORF5 collected from 2012 to 2016, and five deletions in NSP2HVs were detected in PRRSV, in addition to the typical discontinuous deletion of 30 amino acids; thus, this study suggested the approach that should be used for the development of a PRRSV gene-deletion vaccine. Gene-deletion vaccines modify a highly infectious virus into a weakened virus by selectively deleting or mutating genes associated with virulence. This alteration ensures that the vaccine retains its immunogenicity while preventing the virus from easily regaining its virulence in animals. Additionally, it allows for the clear differentiation between natural infection by the wild virus and immunity acquired through vaccination. Currently, gene-deletion vaccines are widely used as PRRSV vaccines, and the corresponding development technology will also help in the research and development of a new PRRSV vaccine.

## 8. Synthetic Peptide Vaccines

Synthetic peptide vaccines are artificially synthesized using chemical synthesis technology based on the amino acid sequence of the viral protein derived from the viral genome sequence [102]. They are usually prepared by coupling them with an adjuvant. Mokhtar et al. [103] used a synthetic peptide library of the whole proteome and repeatedly tested and infected small groups of pigs immunized against PRRSV-1Olot/91, which better demonstrated the response of T cells to PRRSV-1. However, the screening method of the proteome full-domain peptide library and the identified antigen need to be further evaluated in the context of next-generation vaccine development. Mokhtar et al. [104] identified M and NSP5 proteins as conservative targets for multifunctional CD8 and CD4T cells. Peptides representing M and NSP5 were then encapsulated in hydrophobically modified chitosan particles through the incorporation of synthetic poly TLR2/TLR7 agonists and used with adjuvants comprising model B-cell PRRSV antigens. The results showed a significant response to M/NSP5-specific IFN-γ from CD8 T cells; however, future work should focus on enhancing the cross-presentation of M/NSP5 with CD8 T cells. In the development of synthetic peptide vaccines, we should focus on enhancing the cross-presentation of M/NSP5 with CD8 T cells.

As a new type of vaccine, synthetic peptide vaccines have had a late start, but they have good safety and stability and low production costs. They could thus help to rapidly develop effective targeted vaccines according to the epidemic situation, leading to good development prospects [105]. However, synthetic peptide vaccines are produced via artificial synthesis, and the protective ability of a single linear antigen epitope in pigs [106] might not be good, and the immunogenicity is low. Therefore, the connection of multiple neutralizing epitopes in series and the addition of appropriate adjuvants are currently effective means to enhance the immunogenicity of synthetic peptide vaccines, and an in-depth study of the spatial conformation of PRRSV and the immune system in pigs is key to the further development of synthetic peptide vaccines.

## 9. VLP Vaccines

VLPs are hollow protein particles assembled from one or several viral structural proteins that are expressed in vitro. Their appearance is similar to that of natural virus particles, but they contain no nucleic acids, and thus, they cannot multiply and have no pathogenicity. They also have good immunogenicity and can be used at a low immune-inducing dose, with a diameter of approximately 20–150 nm. After immunization, they can induce humoral, cellular, and mucosal immunity [107].

VLP vaccines have been developed commercially to prevent and control other diseases, with sound clinical effects [108]. Researchers have also explored the development of PRRSV VLP vaccines and made specific progress [109]. Previous research has shown that the formation of PRRSV VLPs requires more than two proteins [110]. The immune activity of VLPs containing PRRSV GP5 and M proteins prepared using the baculovirus expression system was not different from that of natural proteins, and mice immunized with different doses of VLPs exhibited specific immune activities [111]. Some studies have shown that GP2b plays an important role in promoting the release of VLPs [112]. An immunogenic gene of PRRSV can also be embedded into the immunogenic genes of other pathogens to synthesize VLPs, which can be used to simultaneously prevent and control multiple pathogens. Wang et al. [113] prepared VLPs that were chimeric with the main immunogenic proteins, SIV and PRRSV. Animal immune tests found that chimeric VLPs could be a potentially safe and effective candidate vaccine for PRRSV and SIV and could induce cellular and humoral immune responses in the body. Kang et al. [112] prepared VLPs composed of the main immunogenic proteins of PCV2, porcine parvovirus, and PRRSV and immunized pigs via intramuscular injection, their drinking water, nasal drip, and other methods to test their immune-inducing effects. The results showed that VLPs could stimulate humoral and cellular immune responses to the three pathogens simultaneously in pigs, confer protection to pigs at all stages, and be used as vaccines against multiple diseases. This has provided a foundation for the development of new vaccines. A fusion protein of the hepatitis B virus core protein and PRRSV GP5 was also expressed using an Escherichia coli expression system, which could be assembled to form VLPs independently. In vitro tests showed that VLPs could inhibit the infection of cells by PRRSV, and a higher concentration of VLPs resulted in a better inhibitory effect [114,115]. VLPs containing PRRSV GP5-GP4-GP3-GP2a-M protein were further prepared using a new baculovirus expression system. After embedding them with PLGA nanoparticles, pigs were immunized via an intranasal drip and then challenged to detect the effect on immunity. The results showed that after immunization, humoral immunity and cellular immunity were simultaneously induced in the pig body, which could significantly reduce the viral load in the pig lung [116]. However, research on PRRS VLP vaccines has just begun. The proteins of PRRSV that can promote the formation of VLPs and maintain good immunogenicity and adjuvants that can improve the immune-inducing effects need to be explored further.

## 10. Potential Effective Vaccines

Adjuvants are crucial for the immune-modulating effects of inactivated vaccines [117]. Commonly used adjuvants mainly include insoluble aluminum salts, mineral oil adjuvants, propolis, microorganisms and their metabolites, immune-stimulating complexes, and cytokines [118]. An intranasal injection strategy using a nanoparticle-encapsulated inactivated PRRSV vaccine and polylactic acid-glycolic acid or *Mycobacterium tuberculosis* whole-cell lysate as adjuvants has been widely used to induce cross-protective anti-PRRSV immunity against heterogeneous PRRSV strains [119,120]. In addition, an intranasal adjuvant composed of *Escherichia coli* and the thermolabile enterotoxin B subunit of ginsenoside Rg1, when added to the inactivated PRRSV vaccine, can upregulate IFN-I signal transduction and enhance the resistance response in mice [121]; however, further research is needed to evaluate the efficacy of this combined adjuvant in conferring immune protection to pigs. Toll-like receptor (TLR) agonist 9 can reduce viremia and strengthen non-antigen-specific IFN-γ levels after vaccination. It is thus expected to become a promising adjuvant in combination with inactivated vaccines [122,123]. Chen et al. [123] used the new PRRSV-specific IgM monoclonal antibody (Mab)-PR5nf1 as the vaccine adjuvant, prepared a mixture of inactivated PRRSV (KIV) and Mab-PR5nf1 with a standard adjuvant to enhance the protection mediated by the PRRSV-KIV vaccine, and further compared its effect on the immune system with that of ordinary KIV and MLV vaccines. The results showed that the addition of PRRSV-specific IgM to the PRRSV-KIV vaccine could significantly improve the overall survival rate and CMI, as determined by serum IFN-γ quantitation and an IFN-γ ELISpot assay. Therefore, this unique formula, combined with the new adjuvant, might enhance the immune response to the inactivated PRRSV vaccine. Recent studies have also shown that the pig intestinal flora can reduce the stress response during vaccination with inactivated vaccines, which could lead to the development of new strategies to improve the efficacy of vaccines [124]. In their study, Maragkakis et al., focused on examining the impact of ID and IM vaccination of PRRSV MLVs on Fas-related cell apoptosis. The findings revealed a significant positive correlation and moderate correlation between the PRRSV viral load and the Fas level. The rise in serum Fas levels among vaccinated pigs suggests the inhibition of cell apoptosis. This study sheds light on the potential role of MLV vaccines from a different perspective [125].

The optimal immunization plan is not only related to the immunization dose and the interval between booster immunization but also an important factor in the vaccination route. The common vaccination methods include intramuscular injection, intradermal injection, and subcutaneous injection. The immune response intensity induced by different vaccination routes also varies [126]. Muscle tissue contains only a few resident immune cells. However, intramuscular injection of vaccines leads to significant recruitment of immune cells and local inflammatory reactions [127]. The skin is rich in a large number of cells with immune functions, participating in the initial stages of inflammation, repair, and immune response. Subcutaneous tissue is mainly composed of adipocytes, with only a few macrophages and lymphocytes [128]. Recent studies have shown that intradermal injection might improve the efficacy of vaccines because it leads to lower levels of IL-10 and IFN-γ-SC [129]. The intestinal microbiota plays a vital role in the efficacy of the PRRS-MLV vaccine, providing new ideas for improving the effectiveness of MLV vaccines [130].

To address the issue of the poor immunogenicity of existing vaccines, researchers have found ways to improve vaccine efficacy. VCSL1-GP5-N33D is a low-glycosylation chimeric virus developed using reverse genetics technology that contains the GP5 outer region of the Korean genealogy-1 wild-type strain. Applying the vCSL1-GP5-N33D inactivated vaccine to pig farms experiencing a PRRSV epidemic could result in high serum virus-neutralizing antibody titers, but no pigs in the negative control showed an SVN antibody titer. The overall results of animal experiments showed that the vCSL1-GP5-N33D inactivated vaccine is a promising candidate [131]. Cui et al. [132] developed a GP5-Mosaic DNA Vaccine Prime/GP5-Mosaic rVaccine for GP5-Mosaic, and experimental results showed that vaccination with GP5-Mosaic-based vaccines resulted in cellular reactivity and higher levels of NAs to both VR2332 and MN184C PRRSV strains. Moreover, viral loads in serum, tissues, PAMs, and bronchoalveolar lavage fluids were significantly lower, and lung lesions were less severe after a challenge with either MN184C or VR2332. The results indicate that GP5-Mosaic vaccines, using a DNA-prime/VACV boost regimen, could protect pigs against heterologous viruses.

## 11. Summary and Outlook

PRRS is one of the most severe diseases in the global pig industry, and it can damage the immune organs of pigs, cause immunosuppression, and increase susceptibility to other diseases. PRRSV is associated with an ADE phenomenon and is prone to mutations. Different viral strains require better cross-protection, making it challenging to develop traditional vaccines. The current inactivated vaccines have a short duration of immune-inducing effects, require large vaccination doses, and have high costs. There is also the risk that the virulence of MLVs will increase. Moreover, identifying whether antibodies are produced in response to wild virus infection or vaccine immunization is difficult. DNA, subunit, or viral vector vaccines have also been tested, but their potential value as substitutes for PRRSV-MLV is still uncertain (Table 3). There are few commercial VLP vaccine products, and many problems need to be solved, such as the selection and optimization of gene sequences, the selection of expression systems, improvements in expression quantity, the large-scale purification of VLPs, reductions in cost, the selection and addition ratio of adjuvants, and the evaluation of vaccine efficacy. Therefore, the development of a new vaccine to prevent and control this disease is urgently needed. In addition, a new hybrid PRRSV strain can be developed using DNA recombination or synthetic PRRSV structural protein gene fragments based on phylogenetic analysis. Furthermore, if the conserved epitopes of NAs can be identified and characterized, the ability of this new hybrid to induce cross-protection against heterogeneous PRRSV can be considered in the design. It is anticipated that PRRSV vaccines designed based on a combination of advanced technologies will show better efficacy and safety than existing vaccines. Overall, the paper suggests that there is a need for continued research and development in the field of PRRS vaccine development to address the limitations of current vaccines and provide better protection against PRRSV.

## Figures and Tables

**Table 2 vetsci-10-00491-t002:** Chinese vaccine statistics.

Source of Vaccine Strains	Vaccine Strain Name	Vaccine Type	Approval Time
CH-1a	KV	Inactivated vaccines	2005
VR-2332	Ingelvac PRRS MLV	MLV	2005
CH-1R	MLV CH-1R	MLV	2007
R98	MLV R98	MLV	2009
HP-PRRSV JXA1	MLV JXA1-R	MLV	2011
HP-PRRSV TJ	MLV TJM-F92	MLV	2011
HP-PRRSV HuN4	MLV HuN4-F112	MLV	2011
HP-PRRSV GD	MLV GDr180	MLV	2015

**Table 3 vetsci-10-00491-t003:** Comparison of the several types of PRRSV vaccines.

Types of Vaccine	Advantage	Disadvantage
MLV vaccine	Strong immunogenicity	Low safety; cannot provide complete protection
Inactivated vaccine	Convenient storage; high security; maternal antibody interference is relatively weak; the vaccine can be used in pigs with weakened immunity	Poor efficacy; multiple vaccinations are needed to strengthen immunity
Subunit vaccine	No active ingredients; high security; the vaccine can be used in pigs with weakened immunity	High production costs; long development time; low immunogenicity
Live vector vaccine	Good immune effect; high-concentration expression of antibodies can be achieved after vaccination; a single dose is sufficient to stimulate long-term protection	Vector viruses may also trigger immune responses, thereby reducing vaccine effectiveness; low production efficiency and high cost
DNA vaccine	The preparation process is simple; the production cycle is short; high security	Low immunogenicity; insufficient resolution of the heterogeneity of PRRSV
Gene-deletion vaccine	Good safety; long immune period, especially suitable for local vaccination	Difficulty in preparation; high cost
Synthetic peptide vaccine	Good safety and stability; low production cost	The protective ability of a single linear antigen epitope may not be good for pigs; low immunogenicity
VLP vaccine	Strong immunogenicity; low immune dose	Difficulty in preparation; high cost

## Data Availability

All datasets are available in the main manuscript. The dataset supporting the conclusions of this article is included within the article.

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
