# Peer review of "Research Progress on the Development of Porcine Reproductive and Respiratory Syndrome Vaccines"

_vetsci, 2023, doi:10.3390/vetsci10080491_

Round 1

Reviewer 1 Report

Most of my comments have been addressed appropriately.  I believe it will be much cleared to the readers.

Much better now.

Author Response

We appreciate the reviewer's affirmation of our manuscript.

Reviewer 2 Report

Comments for the authors

Major comments

-        Table 1: you should take into consideration for the MLV vaccines and the results of the study ‘’ Heterologous Challenge with PRRSV-1 MLV in Pregnant Vaccinated Gilts: Potential Risk on Health and Immunity of Piglets. Animals (Basel). 2022 Feb 12;12(4):450.’’

-        L154-160: add the number of the reference ‘’ Bonckaert et al.’’

-        L146-194: you should take into consideration the results of more recent studies 

·       Evaluation of Intradermal PRRSV MLV Vaccination of Suckling Piglets on Health and Performance Parameters under Field Conditions. Animals (Basel). 2022 Dec 23;13(1):61.

-        L217-235: you should report the results of more studies, such as 

·       Elimination of porcine reproductive and respiratory syndrome virus infection using an inactivated vaccine in combination with a roll-over method in a Hungarian large-scale pig herd. Acta Vet Scand. 2022 May 7;64(1):12

·       The use of a whole inactivated PRRS virus vaccine administered in sows and impact on maternally derived immunity and timing of PRRS virus infection in piglets. Vet Rec Open. 2022 Apr 5;9(1):e34.

·       Maternal Autogenous Inactivated Virus Vaccination Boosts Immunity to PRRSV in Piglets. Vaccines (Basel). 2021 Jan 31;9(2):106.

-L485-497: you should report the results of more studies, such as 

·       Investigation of Fas (APO-1)-Related Apoptosis in Piglets Intradermally or Intramuscularly Vaccinated with a Commercial PRRSV MLV. Viral Immunol. 2022 Mar;35(2):129-137.

Minor comments

-        L 80, 241….: … in vivo (italics)

-        L85: After the passive transfer..

-        L 82, 422….: .. in vitro (italics)

-        L98: .. CH-1a-like

-        L101: . HP-PRRSV-like

-        L247: . in virus-infected cells

-        L248: .. associate with non-covalent

-        L292: … has a broad application..

-        L330: .. PRRSV-specific antibodies

-        L351: .. Compared with that traditional

Author Response

Table 1: you should take into consideration for the MLV vaccines and the results of the study “Heterologous Challenge with PRRSV-1 MLV in Pregnant Vaccinated Gilts: Potential Risk on Health and Immunity of Piglets. Animals (Basel). 2022 Feb 12;12(4):450.”

Response: We have added some contents in Table 1 based on the research results.

L154-160: add the number of the reference “Bonckaert et al.”

Response: We have added the number of the reference “Bonckaert et al.”, it is in line 159.

L146-194: you should take into consideration the results of more recent studies “Evaluation of Intradermal PRRSV MLV Vaccination of Suckling Piglets on Health and Performance Parameters under Field Conditions. Animals (Basel). 2022 Dec 23;13(1):61.”

Response: The following contents have been added. They are in lines 170-174.

Maragkakis et al. [45] conducted a study wherein they observed that the intradermal (ID) injection of PRRSV-1 MLV vaccine in piglets had a beneficial effect on their health and production performance, as opposed to the intramuscular (IM) inoculation. This positive impact was evidenced by an augmentation in body weight at slaughter and a reduction in lung and pleurisy lesion scores.

L217-235: you should report the results of more studies, such as “Elimination of porcine reproductive and respiratory syndrome virus infection using an inactivated vaccine in combination with a roll-over method in a Hungarian large-scale pig herd. Acta Vet Scand. 2022 May 7;64(1):12” “The use of a whole inactivated PRRS virus vaccine administered in sows and impact on maternally derived immunity and timing of PRRS virus infection in piglets. Vet Rec Open. 2022 Apr 5;9(1):e34.” “Maternal Autogenous Inactivated Virus Vaccination Boosts Immunity to PRRSV in Piglets. Vaccines (Basel). 2021 Jan 31;9(2):106.”

Response: The following contents have been added. They are in lines 245-259.

The successful implementation of Hungary's PRRS Eradication Programme has been substantiated by recent research. This achievement was accomplished by administering an inactivated vaccine to sows and adopting segregated rearing for their offspring. Notably, the production remained nearly uninterrupted throughout the entirety of the population replacement process. Several studies have indicated that the utilization of a combination of MLV vaccine and inactivated vaccine provides enhanced protection against PRRSV infection in pigs. The administration of the MLV vaccine during the eighth week of gestation in sows, followed by a re-inoculation of the commercial inactivated vaccine three weeks prior to delivery, effectively diminishes the occurrence rate of PRRSV. Furthermore, it has been noted that weaned piglets exhibit a greater proportion of PRRSV seropositive individuals. Furthermore, the combination of MLV vaccine and inactivated vaccine can provide partial protection to piglets, reducing lung pathological loss, improving weight gain, and decreasing the viremia. This effect may be attributed to the production of NAs in sows induced by the inactivated vaccine.

L485-497: you should report the results of more studies, such as “Investigation of Fas (APO-1)-Related Apoptosis in Piglets Intradermally or Intramuscularly Vaccinated with a Commercial PRRSV MLV. Viral Immunol. 2022 Mar;35(2):129-137.”

Response: The following contents have been added. They are in lines 518-523.

In their study, Maragkakis et al. focused on examining the impact of ID and IM vaccination of PRRSV MLVs on Fas-related cell apoptosis. The findings revealed a significant positive correlation and moderate correlation between PRRSV viral load and Fas level. The rise in serum Fas levels among vaccinated pigs suggests the inhibition of cell apoptosis. This study sheds light on the potential role of MLV vaccines from a different perspective.

Minor comments

L80, 241….: … in vivo (italics)

L85: After the passive transfer..

L82, 422….: .. in vitro (italics)

L98: .. CH-1a-like

L101: . HP-PRRSV-like

L247: . in virus-infected cells

L248: .. associate with non-covalent

L292: … has a broad application..

L330: .. PRRSV-specific antibodies

L351: .. Compared with that traditional

Response: We have already italicized “in vivo” and “in vitro”, they are in lines 79, 82, 228, and 454. We have changed “After passive transfer..” to “After the passive transfer..”, it is in line 85. We have corrected the spelling of some words, they are in lines 98, 101, 274, and 359. We have changed “associate by non-covalent” to “associate with non-covalent”, it is in line 275. We have changed “has broad application” to “has a broad application”, it is in line 320. We have changed “Compared with that of traditional” to “Compared with that traditional”, it is in line 381.

Reviewer 3 Report

Minor Comments

This research article investigates provided information does not explicitly state what data has been used in this paper. However, it can be assumed that the authors have reviewed and analyzed existing literature and research studies on the development of porcine reproductive and respiratory syndrome (PRRS) vaccines. The paper summarizes the advantages, disadvantages, and applicability of various PRRSV vaccines, including modified live virus, inactivated, recombinant subunit, live vector, DNA, gene-deletion, synthetic peptide, and virus-like particle vaccines, as well as various other vaccines. The manuscript is well written, and the experimental design/data analysis are robust.  I would recommend the following major/minor comments to the authors.

Point 1: The limitations of this paper are not explicitly stated in the provided information. However, it is important to note that the information presented in the paper is focused on the development of porcine reproductive and respiratory syndrome (PRRS) vaccines in China and may not be applicable to other regions or countries. Additionally, the paper may not include the most recent developments in PRRS vaccine research as it was published in 2020.

Point 2: What are the safety concerns associated with PRRSV-MLVs?

Point 3:  What is the significance of gene-deletion vaccines in PRRS prevention?

Point 4: Further evaluation of the screening method of the proteome full-domain peptide library and the identified antigen in the context of next-generation vaccine development.

  • Focusing on enhancing the cross-presentation of M/NSP5 with CD8 T cells in the development of synthetic peptide vaccines.
  • Developing adjuvants with beneficial effects to enhance the specific immune response to PRRSV subunit vaccines.
  • Continuing research and development of genetically engineered vaccines and marker vaccines to address the limitations of current PRRS vaccines.
  • Developing new vaccines that can better protect against PRRSV in China, particularly against heterologous strains and strains with higher genetic variation and recombination rates.

Point 5: Overall, the paper suggests that there is a need for continued research and development in the field of PRRS vaccine development to address the limitations of current vaccines and provide better protection against PRRSV.

      Point 6: Minor English correction required in the revised version.

        Good Luck

Minor English correction required in the revised version.

Author Response

Point 1: The limitations of this paper are not explicitly stated in the provided information. However, it is important to note that the information presented in the paper is focused on the development of porcine reproductive and respiratory syndrome (PRRS) vaccines in China and may not be applicable to other regions or countries. Additionally, the paper may not include the most recent developments in PRRS vaccine research as it was published in 2020.

Response: We have made modifications according to your suggestion. We have added the PRRS eradication program in Hungary which has been successfully implemented. They are in lines 245-259. We have added the recent developments in PRRS vaccine research as it was published in 2020. Recent research has revealed that a new strain in Danish resulted from a recombination event involving the Amervac strain (Unistrain PRRS vaccine; Hipra) and the 96V198 strain (Suvaxyn PRRS; Zoetis AH). The 96V198 strain primarily contributed to the genetic makeup of the strain, encompassing ORFs 1-2 and a portion of ORF 3. On the other hand, the Amervac strain acted as the minor parent, contributing to the remaining portion of the genome. They are in lines 187-192.

Point 2: What are the safety concerns associated with PRRSV-MLVs?

Response: The challenge of safety arises for PRRSV-MLVs due to vertical transmission, a decrease in the number of piglets born alive, and the potential to cause persistent infection in vaccinated hosts. They are in lines 177-179.

Point 3: What is the significance of gene-deletion vaccines in PRRS prevention?

Response: The gene-deletion vaccine modifies a highly infectious virus into a weakened virus by selectively deleting or mutating genes associated with virulence. This alteration ensures that the vaccine retains its immunogenicity while preventing the virus from easily regaining its virulence in animals. Additionally, it allows for a clear differentiation between natural infection by the wild virus and immunity acquired through vaccination. They are in lines 416-421.

Point 4: Further evaluation of the screening method of the proteome full-domain peptide library and the identified antigen in the context of next-generation vaccine development.

Focusing on enhancing the cross-presentation of M/NSP5 with CD8 T cells in the development of synthetic peptide vaccines.

Developing adjuvants with beneficial effects to enhance the specific immune response to PRRSV subunit vaccines.

Continuing research and development of genetically engineered vaccines and marker vaccines to address the limitations of current PRRS vaccines.

Developing new vaccines that can better protect against PRRSV in China, particularly against heterologous strains and strains with higher genetic variation and recombination rates.

Response: We agree with these viewpoints and add them in lines 111-113、118-122、326-327、439-440.

Continuing research and development of genetically engineered vaccines and marker vaccines can address the limitations of current PRRS vaccines.

In the context of next-generation vaccine development, further evaluation of the screening method of the proteome full-domain peptide library and the identified antigens play an important role in developing new vaccines that can better protect against PRRSV in China, particularly against heterologous strains and strains with higher genetic variation and recombination rates.

Therefore, developing adjuvants with beneficial effects will help to enhance the specific immune response to PRRSV subunit vaccines.

In the development of synthetic peptide vaccine, we should focus on enhancing the cross-presentation of M/NSP5 with CD8 T cells.

Point 5: Overall, the paper suggests that there is a need for continued research and development in the field of PRRS vaccine development to address the limitations of current vaccines and provide better protection against PRRSV.

Response: We agree with this viewpoint and add them in lines 566-569.

Overall, the paper suggests that there is a need for continued research and development in the field of PRRS vaccine development to address the limitations of current vaccines and provide better protection against PRRSV.

Point 6: Minor English correction required in the revised version.

Response: We have further carefully revised the writing of this manuscript.

We have already italicized “in vivo” and “in vitro”, they are in lines 79, 82, 228, and 454. We have changed “After passive transfer..” to “After the passive transfer..”, it is in line 85. We have corrected the spelling of some words, they are in lines 98, 101, 274, and 359. We have changed “associate by non-covalent” to “associate with non-covalent”, it is in line 275. We have changed “has broad application” to “has a broad application”, it is in line 320. We have changed “Compared with that of traditional” to “Compared with that traditional”, it is in line 381.

Reviewer 4 Report

Brief summary

The review titled “Research Progress on the Development of Porcine Reproductive and Respiratory Syndrome Vaccines” by Zhang et al. aims to provide a theoretical basis for PRRS vaccine research and development that protect against respiratory and reproductive disease in the pig industry. The main contribution is to describe the different available vaccines against this disease, which is a valuable source of information for researchers who are working in this field. Furthermore, this review highlights the advantages and disadvantage of currently available commercial vaccines including their immunogenicity, safety, and production costs, among other factors.

General concept comments

This review article is relevant for the development of vaccines against porcine reproductive and respiratory syndrome. Its information is of interest to researchers working in the field of vaccines against swine diseases, both viral and bacterial.

The authors have carried out a good literature search with 113 references (although in the manuscript they are numbered up to 112) covering the period from 1988 to 2023 (12 references up to 1999, 73 references from 2000 to 2019 and 28 from 2020 to 2023).

However, the manuscript needs a mayor revision as many references do not match their numbering in the References section with those written in the text of the manuscript (see specific comments). These errors suggest that it is necessary to check whether the information written throughout the manuscript corresponds to the number of each reference.

The manuscript is in a well-structured manner. The tables are appropriate, although some changes are necessary to improve their viewing and interpretation (see specific comments). I also suggest adding more tables that summarize the information described for each vaccine type. This would make it easier to visualize the different studies carried out in each section (subunit, live vector, DNA,... vaccines). If it is decided to incorporate new tables, these may include the characteristics of the vaccines, their advantages/disadvantages, the type of cellular/humoral immune response... for each study conducted by other researchers.

This article is an interesting study but needed of mayor modifications (mainly in references in text and Reference section) for the definitive version.

Specific comments 

Abstract:

Line 32: “prevention of PRRS”. The authors should add “prevention and/or control”.

Introduction:

Lines 37 to 41; 99 to 111: Literature references should be added to support the information.

Lines 116-117: The goal described by the authors only refers to “the development of new vaccines to protect against PRRSV in China”. I think it should be extended to all countries where this disease is diagnosed.

“2. MLV vaccines” section:

Lines 134 to 138: Literature references should be added to support the information.

Line 156: Add the reference number: Bonckaert et al. [29]

Line 161. Is the expression “…so. [29]” ok?

Line 164: Which trimester of pregnancy? Late-term?

Line 171: “Charerntantanakul and Wang et al.” is not right. This reference is Charerntantanakul [38].

Lines 171 to 173: The authors must make clearer if the information located in these lines was described by [38], [35-44]

Lines 186 to 189: It is not clear if the information written between these lines was described by [50]?

“3. Inactivated vaccine” section

Line 199: The authors should change “at home and abroad” for “at China and other countries”.

Line 201: The reference of Opriessnig et al. is number 54 in the Reference section and not number 53 as written by the authors in the text.

Line 203: "Nielsen" is not right. According to the References section, the author's name is Nilubol et al. [55] and the number 54 written in the text of the manuscript is not right.

Lines 201 to 211. The authors must check that the information corresponds to the references which they have written.

Lines 221 to 224 and 230 to 231: The authors must make it clearer which bibliographic references correspond to the information.

“4. Subunit vaccine” section

Lines 237 to 240; 249 to 254. Literature references should be added to support the information.

Lines 256 to 280: The reference number written in the text of the manuscript does not coincide with the number in the References section. The authors need to revise through section for reference numbering and its correspondence with the content. E.g.,

·       Hu et al. is 70, but not 69

·       An et al. is 71, but not 70

Lines 254 to 283: I suggest adding a table with the information of all the studies/articles mentioned in this section to better visualize the most important characteristics of each of them. If the table is included, the text must be rewritten so as not to repeat information.

“5. Live vector vaccines” section

Lines 300 to 302: Literature references should be added to support the information.

Lines 303 to 304: add bibliographic references for each type of vector mentioned in the text.

Lines 305 to 317: The reference number written in the text of the manuscript does not coincide with the number in the References section. E.g.,

·       Alonso et al. is 76 but not 75

·       Fang et al. is 77, but not 76

·       Tian et al. 78, but not 77

“6. DNA vaccines” section

Lines 326 to 329 and 359 to 363: bibliographic references should be added to support the information.

Lines 329 to 348: the reference numbers do not coincide with the number in Reference section. The authors must check that the information corresponds to the bibliographic references.

I suggest adding a table with the information of all the studies/articles mentioned in this section to better visualize the most relevant features of each of them. If the table is included, the text must be rewritten so as not to repeat information.

“7. Gene-deletion vaccines” section

The authors must make modifications like those mentioned in the sections above.

Lines 365 to 366. Add a reference to support the information given in the first sentence.

Lines 366 to 383. The reference numbers do not match with the number in Reference section. The authors must check that the information corresponds to the references.

If possible, add a table with the most relevant information on this type of vaccines. This would be a summary of the different studies carried out.

“8. Synthetic Peptide vaccines” section

Lines 395 to 397. Add a reference to support the information given in the first sentence.

Lines 409 to 414. Add bibliographic references to support the information.

Line 398: Mokhtar et al. is 91 but not 90.

If possible, summarize the most important data from each study conducted on this type of vaccine in a table.

“9. Virus-like particle vaccines” section

This section requires modifications like those described in the previous sections.

Lines 421 to 426 and 429 to 430. Add bibliographic references to support the information.

The authors must check that the numbering of references in the text matches those in the References section.

“10. Poten Effective vaccine” section

Line 460. Add a reference to support the information given in the first sentence.

Lines 463 and 469: the reference numbers do not coincide with the number in Reference section. The authors must check that the information corresponds to the references included.

Be careful in the References section because there are two 101 numbers.

 Lines 464 and 467: Mycobacterium tuberculosis and Escherichia coli in italics.

Line 473: Change “Chen [105] and others” for “Chen et al. [106].

Line 479: Add what "CMI" means.

Lines 485 to 489: Add literature references to support the information.

Lines 491 to 494. Does all the information described in these lines correspond to the same reference [108]?

Line 505: Cui et al. [112]

Line 527: The reference to Table 3 in the text should be moved. As it stands in the current version, it seems that the table only has information on VLP vaccines.

Tables

Table 1.

The title of the table should include the word MLV. E.g., Foreign MLV vaccine statistics.

The PRRSV1 vaccines should be ordered by approve time. Thus, the 1994 vaccine would be placed first, then the 2012, 2015 and the last one would be the vaccine of 2018.

The information in the columns "Vaccine efficacies" and "Immune responses" should start with a capital letter. E.g.,

·       Increases in average dairy wight gain; reduces ....

·       Reduces viremia.

·       Humoral and cellular immune responses

·       Cellular immune responses....

The rows should be spaced a little further apart from each other so that the information that corresponds to each row can be better displayed.

Table 2:

In the title, the authors should change the word "Domestic" for "Chinese".

Table 3:

Change the title for “Comparison of the several types of PRRSV vaccines.”

The rows should be spaced a little further apart from each other so that the information that corresponds to each row can be better displayed.

Reference section

Reference 15 is incomplete, pages are missing.

Reference 87, the Chinese characters that appear in this reference should be deleted.

There are two references with the number of 101.

Author Response

Dear reviewer 4:

We thank the reviewer for the constructive comments and suggestions to improve our manuscript. We agreed to most of the suggestions and comments. Our point-by-point responses are included as the following.

Abstract:

Line 32: “prevention of PRRS”. The authors should add “prevention and/or control”.

Response: We have added “prevention and control”, it is in line 33.

Introduction:

Lines 37 to 41; 99 to 111: Literature references should be added to support the information.

Response: We have added references to support the information, they are in lines 42, 100, 102, and 108.

Lines 116-117: The goal described by the authors only refers to “the development of new vaccines to protect against PRRSV in China”. I think it should be extended to all countries where this disease is diagnosed.

Response: We have made the modifications according to your instructions, they are in lines 118-119.

The goal is to provide a theoretical foundation for the development of new vaccines that can offer enhanced protection against PRRSV in regions affected by this disease.

“MLV vaccines” section:

Lines 134 to 138: Literature references should be added to support the information.

Response: We have added reference to support the information, it is in line 139.

Line 156: Add the reference number: Bonckaert et al. [29]

Response: We have added reference number, it is in line 161.

Line 161. Is the expression “…so. [29]” ok?

Response: We have rewritten the sentence, they are in lines 165-167.

Compared to the control group, the duration of viremia in the vaccine group showed a slight decrease, although the difference was not statistically significant.

Line 164: Which trimester of pregnancy? Late-term?

Response: We have corrected this sentence, they are in lines 168-170.

Further, MLV vaccines against PRRSV-2 have been shown to offer cross-protection against heterologous PRRSV-1 in pregnant gilts throughout their trimester.

Line 171: “Charerntantanakul and Wang et al.” is not right. This reference is Charerntantanakul [38].

Response: We have added reference from Wang et al., they are in line 183.

Lines 171 to 173: The authors must make clearer if the information located in these lines was described by [38], [35-44]

Response: The references we cited are [35,44], instead of [35-44], it is in line 183.

Lines 186 to 189: It is not clear if the information written between these lines was described by [50]?

Response: The information written between lines 199-201 was described by [56], it is in line 201.

“Inactivated vaccine” section

Line 199: The authors should change “at home and abroad” for “at China and other countries”.

Response: We have changed “at home and abroad” to “at China and other countries”, they are in lines 214-215.

Line 201: The reference of Opriessnig et al. is number 54 in the Reference section and not number 53 as written by the authors in the text.

Response: We have made overall adjustments to the references in the manuscript, it is in line 218.

Line 203: "Nielsen" is not right. According to the References section, the author's name is Nilubol et al. [55] and the number 54 written in the text of the manuscript is not right.

Response: We have changed “Nielsen” to “Nilubol” and made overall adjustments to the references in the manuscript, it is in line 218.

Lines 201 to 211. The authors must check that the information corresponds to the references which they have written.

Response: We have made overall adjustments to the references in the manuscript, they are in lines 220, 225, and 226.

Lines 221 to 224 and 230 to 231: The authors must make it clearer which bibliographic references correspond to the information.

Response: We have made overall adjustments to the references in the manuscript, they are in lines 234, 236, 241, and 245.

“Subunit vaccine” section

Lines 237 to 240; 249 to 254. Literature references should be added to support the information.

Response: We have added reference to support the information, they are in lines 268 and 271.

Lines 256 to 280: The reference number written in the text of the manuscript does not coincide with the number in the References section. The authors need to revise through section for reference numbering and its correspondence with the content. E.g.,Hu et al. is 70, but not 69; An et al. is 71, but not 70

Response: We have made overall adjustments to the references in the manuscript. They are in lines 301 and 305.

Lines 254 to 283: I suggest adding a table with the information of all the studies/articles mentioned in this section to better visualize the most important characteristics of each of them. If the table is included, the text must be rewritten so as not to repeat information.

Response: We agree with your point of view, but adding a table may pose challenges for this manuscript in terms of type setting. We believe that the current style has effectively conveyed the intended message to readers.

“Live vector vaccines” section

Lines 300 to 302: Literature references should be added to support the information.

Response: We have added reference to support the information, it is in line 334.

Lines 303 to 304: add bibliographic references for each type of vector mentioned in the text.

Response: We have added reference for each type of vector mentioned in the text, it is in line 336.

Lines 305 to 317: The reference number written in the text of the manuscript does not coincide with the number in the References section. E.g., Alonso et al. is 76 but not 75; Fang et al. is 77, but not 76; Tian et al. 78, but not 77

Response: We have made overall adjustments to the references in the manuscript, they are in lines 336, 338, and 341.

“DNA vaccines” section

Lines 326 to 329 and 359 to 363: bibliographic references should be added to support the information.

Response: We have added reference to support the information, they are in lines 360 and 394.

Lines 329 to 348: the reference numbers do not coincide with the number in Reference section. The authors must check that the information corresponds to the bibliographic references.

Response: We have made overall adjustments to the references in the manuscript, they are in lines 360, 376, and 379.

I suggest adding a table with the information of all the studies/articles mentioned in this section to better visualize the most relevant features of each of them. If the table is included, the text must be rewritten so as not to repeat information.

Response: We agree with your point of view, but adding a table may pose challenges for this manuscript in terms of type setting. We believe that the current style has effectively conveyed the intended message to readers.

“Gene-deletion vaccines” section

The authors must make modifications like those mentioned in the sections above.

Lines 365 to 366. Add a reference to support the information given in the first sentence.

Response: We have added a reference to support the information given in the first sentence, it is in line 397.

Lines 366 to 383. The reference numbers do not match with the number in Reference section. The authors must check that the information corresponds to the references.

Response: We have made overall adjustments to the references in the manuscript, they are in lines 397, 399, 408, and 414.

If possible, add a table with the most relevant information on this type of vaccines. This would be a summary of the different studies carried out.

Response: We agree with your point of view, but adding a table may pose challenges for this manuscript in terms of type setting. We believe that the current style has effectively conveyed the intended message to readers.

“Synthetic Peptide vaccines” section

Lines 395 to 397. Add a reference to support the information given in the first sentence.

Response: We have added a reference to support the information given in the first sentence, it is in line 429.

Lines 409 to 414. Add bibliographic references to support the information.

Response: We have added reference to support the information, they are in line 446 and 447.

Line 398: Mokhtar et al. is 91 but not 90.

Response: We have made overall adjustments to the references in the manuscript, it is in line 430.

If possible, summarize the most important data from each study conducted on this type of vaccine in a table.

Response: We agree with your point of view, but adding a table may pose challenges for this manuscript in terms of type setting. We believe that the current style has effectively conveyed the intended message to readers.

“Virus-like particle vaccines” section

This section requires modifications like those described in the previous sections.

Lines 421 to 426 and 429 to 430. Add bibliographic references to support the information.

Response: We have added reference to support the information, they are in line 460 and 464.

The authors must check that the numbering of references in the text matches those in the References section.

Response: We have made overall adjustments to the references in the manuscript.

“Poten Effective vaccine” section

Line 460. Add a reference to support the information given in the first sentence.

Response: We have added a reference to support the information given in the first sentence, it is in line 496.

Lines 463 and 469: the reference numbers do not coincide with the number in Reference section. The authors must check that the information corresponds to the references included.

Response: We have reviewed this section and the corresponding references, and made modifications. It is in lines 498.

Be careful in the References section because there are two 101 numbers.

Response: We have made comprehensive revisions to the references.

Lines 464 and 467: Mycobacterium tuberculosis and Escherichia coli in italics.

Response: We have already italicized “Mycobacterium tuberculosis” and “Escherichia coli”, they are in lines 500 and 502.

Line 473: Change “Chen [105] and others” for “Chen et al. [106].

Response: We have changed “Chen [105] and others” to “Chen et al. [112]”, it is in line 509.

Line 479: Add what "CMI" means.

Response: "CMI" means “cell-mediated immune”, it has been explained in line 226.

Lines 485 to 489: Add literature references to support the information.

Response: We have added a reference to support the information given in the first sentence, it is in line 529.

Lines 491 to 494. Does all the information described in these lines correspond to the same reference [108]?

Response: Yes, it is, and the information described in these lines now correspond to the reference [127]. It is in line 534.

Line 505: Cui et al. [112]

Response: We have made overall adjustments to the references in the manuscript, it is in line 545.

Line 527: The reference to Table 3 in the text should be moved. As it stands in the current version, it seems that the table only has information on VLP vaccines.

Response: We have deleted “(Table 3)” in the text.

Tables

Table 1.

The title of the table should include the word MLV. E.g., Foreign MLV vaccine statistics.

Response: We have changed the title to "Foreign MLV vaccine statistics.", it is in line 150.

The PRRSV1 vaccines should be ordered by approve time. Thus, the 1994 vaccine would be placed first, then the 2012, 2015 and the last one would be the vaccine of 2018.

Response: We have modified the order according to your request.

The information in the columns "Vaccine efficacies" and "Immune responses" should start with a capital letter. E.g., Increases in average dairy wight gain; reduces ....; Reduces viremia.; Humoral and cellular immune responses; Cellular immune responses....

Response: We have made the modifications according to your request.

The rows should be spaced a little further apart from each other so that the information that corresponds to each row can be better displayed.

Response: We have increased the spacing between the rows.

Table 2:

In the title, the authors should change the word "Domestic" for "Chinese".

Response: We have changed "Domestic" to "Chinese", it is in line 151.

Table 3:

Change the title for “Comparison of the several types of PRRSV vaccines.”

Response: We have changed the title to "Comparison of the several types of PRRSV vaccines.", it is in line 578.

The rows should be spaced a little further apart from each other so that the information that corresponds to each row can be better displayed.

Response: We have increased the spacing between the rows.

Reference section

Reference 15 is incomplete, pages are missing.

Response: We have added the pages of Reference 15, it is in line 628.

Reference 87, the Chinese characters that appear in this reference should be deleted.

Response: We have deleted the Chinese characters.

There are two references with the number of 101.

Response: We have made overall adjustments to the references in the manuscript

Best wishes!